# *E*/*Z* Molecular Photoswitches Activated by Two-Photon Absorption: Comparison between Different Families

**DOI:** 10.3390/molecules26237379

**Published:** 2021-12-05

**Authors:** Marco Marazzi, Cristina García-Iriepa, Carlos Benitez-Martin, Francisco Najera, Antonio Monari, Diego Sampedro

**Affiliations:** 1Departamento de Química Analítica, Química Física e Ingeniería Química, Universidad de Alcalá, Ctra. Madrid-Barcelona, Km 33.600, 28805 Alcalá de Henares, Spain; 2Instituto de Investigación Química “Andrés M. del Rio” (IQAR), Universidad de Alcalá, Ctra. Madrid-Barcelona, Km 33.600, 28805 Alcalá de Henares, Spain; 3Departamento de Química Orgánica, Universidad de Málaga-IBIMA, 29071 Málaga, Spain; cbm@uma.es (C.B.-M.); najera@uma.es (F.N.); 4Centro Andaluz de Nanomedicina y Biotecnología (BIONAND), Parque Tecnológico de Andalucía, 29590 Málaga, Spain; 5LPCT, Université de Lorraine and CNRS, F-54000 Nancy, France; antonio.monari@u-paris.fr; 6ITODYS, Université de Paris and CNRS, F-75006 Paris, France; 7Departamento de Química, Centro de Investigación en Síntesis Química, Universidad de La Rioja, Madre de Dios, 53, 26006 Logroño, Spain

**Keywords:** *E*/*Z* photoswitches, photoisomerization, two-photon absorption

## Abstract

Nonlinear optical techniques as two-photon absorption (TPA) have raised relevant interest within the last years due to the capability to excite chromophores with photons of wavelength equal to only half of the corresponding one-photon absorption energy. At the same time, its probability being proportional to the square of the light source intensity, it allows a better spatial control of the light-induced phenomenon. Although a consistent number of experimental studies focus on increasing the TPA cross section, very few of them are devoted to the study of photochemical phenomena induced by TPA. Here, we show a design strategy to find suitable *E*/*Z* photoswitches that can be activated by TPA. A theoretical approach is followed to predict the TPA cross sections related to different excited states of various photoswitches’ families, finally concluding that protonated Schiff-bases (retinal)-like photoswitches outperform compared to the others. The donor-acceptor substitution effect is therefore rationalized for the successful TPA activatable photoswitch, in order to maximize its properties, finally also forecasting a possible application in optogenetics. Some experimental measurements are also carried out to support our conclusions.

## 1. Introduction

The interest on light activated chemical reactions and processes has risen in the last decades, mainly motivated by the clear advantages of light as an external agent over chemical, thermal, or electrochemical stimuli. In particular, light can be easily and precisely switched off/on to control the progress of the reaction, and no waste products (or negligible amounts) are generated. Moreover, solar light can be exploited as a renewable energy source, while the lasers used for irradiation present high spatial and temporal resolution. These facts have led to a wide variety of photochemical applications, ranging from biology to material sciences.

Scientists have focused their efforts not only on discovering novel photochemical reactions, but also on taking advantage of the ones already known to build useful building blocks [1,2,3,4,5]. Photoactive molecular devices, such as molecular motors, rotors, or switches, are a clear example [6,7,8,9,10,11,12,13]. The relevance of this field in the last decade has been acknowledged by the Nobel Prize in Chemistry 2016 awarded to Jean-Pierre Sauvage, Sir J. Fraser Stoddart and Bernard L. Feringa “for the design and synthesis of molecular machines.” Among photoactive molecular devices, molecular switches are the most widely studied, due to their simpler photoactive mechanism and easier implementation in larger systems for applications [6,11,13].

A photoswitch is a molecule that can be interconverted reversibly between two different states by the action of light. Different families of molecular photoswitches have been reported, depending on the photochemical reaction inducing the interconversion, with *E*/*Z* isomerization and electrocyclization being the most common. In particular, a wide variety of *E*/*Z* molecular switches have been reported such as azobenzene [14,15], stilbene [16,17], spiropyrans [18,19], indigoids [20,21], and retinal-like [22,23,24,25,26] or other switches based on natural photoactive chromophores [27,28], such as the chromophore of the green fluorescent protein (GFP) [29,30]. The synthesis of such a considerable amount of *E*/*Z* molecular switches is explained by their widespread use, including in smart materials, such as memory devices, data storage, photocages, photoactive assembled monolayers, polymers, liquid crystals, etc. [6,31,32,33]. The catalog of amazing applications of molecular photoswitches includes, among others, optical switches [34], 3D optical storage [35], and surface relief grafting [36]. In this sense, azobenzenes have been extensively used, even in the context of nonlinear optical properties [37]. Furthermore, biological systems composed of photochromic bio-active small molecules, and larger photoactive peptides/proteins or nucleic acids have also been reported [14,38,39,40,41]. Ideally, the required photon energy responsible for the *E* → *Z* photoisomerization should be different from the one responsible of the backward process, i.e., *Z* → *E* photoisomerization, in order to fully control the switching state and process (Figure 1).

Although molecular photoswitches have been widely studied and applied, they present an evident practical limit, which is their usual activation through high-energy light, generally necessitating absorption in the near-UV or (green to violet) visible window [11]. This prevents a deep penetration of light across the surface of the material or tissue. It is especially problematic in the case of biological/biomedical applications, due to the inherent cytotoxicity of the UV irradiation, which has been associated with skin cancer development [42]. In order to avoid such drawbacks, irradiation with red and near-infrared (NIR) light is therefore highly desirable, since it results in a decrease of the incoming absorption by the material or biological tissue (i.e., an increase of the light penetration depth) and, at the same time, it decreases photodamage [43]. 

Two main strategies have been followed to red-shift the absorption of molecular switches up to the NIR window. The first one involves the chemical modification of the molecular photoswitch in order to extend the π-conjugation [21,44,45,46]. Although some molecular switches absorbing in the NIR limit have been reported, care must be taken when changing the chemical structure of the molecular switch, as it can have a crucial effect on its photochemistry, not only modifying the photoactivated reaction, but also its quantum yield and kinetics. Another strategy that has been implemented is the exploitation of two-photon absorption (TPA) [47,48], in contrast to the more conventional one-photon absorption (OPA). TPA is a non-linear optical process leading to photoexcitation of a given chromophore (in this study, corresponding to a switch) by the simultaneous absorption of two photons: the fundamental requirement of TPA is that the sum of both photon energies matches the vertical transition from the ground state to the excited state. This can be achieved either with two photons having half of the transition energy (degenerate TPA, see Figure 1), or by any combination of photon energies whose sum corresponds to the transition energy (non-degenerate TPA) [49]. Obviously, this allows to considerably shift the incoming photon energy toward the red, inducing a bathochromic shift that could possibly allow to enter the NIR window (ca. 650–1350 nm) [50,51].

Adding up to the energetic considerations, other factors that have an impact on the spatial precision and selectivity of the switching should be taken into account for precise manipulations. This leads to a further advance of the TPA strategy, since the probability that the two photons are absorbed simultaneously is proportional to the square of the light source intensity. Hence, TPA decreases outside of the laser focal point much faster and sharper than OPA. This aspect favors applications where precision is required, as photodynamic therapies for lesions situated in critical organs [52], or multiphoton lithography [53].

From the experimental point of view, it should be noted that the setup needed to activate a photoswitch through TPA is more complex than that usually required for OPA. Moreover, degenerate TPA is usually easier to afford than non-degenerate TPA. In any case, the preservation of the photochemical properties of the switch are ensured, provided that the same electronic excited state is populated.

For these reasons, the design of efficient *E*/*Z* photoswitches with a relatively high TPA cross section is highly desirable for practical purposes. Diverse studies have been already published focused on the TPA properties of different photoswitches families, such as azobenzene [54,55,56,57,58], stilbene [56,59,60], fluorescent protein chromophores [61,62,63], and retinal [64,65,66]. In addition, related compounds have been also explored under TPA conditions: retinal (chromophore of photoactive proteins) [64], retinoic acid derivatives [67], carotenoids and other pigments [68], and several caged compounds [69]. Nevertheless, the reported photoswitches generally present: (i) low TPA cross-section for the excited state responsible of the photoisomerization and (ii) relatively high TPA cross-sections for higher energy excited states, which, in most cases, are not driving photoisomerization. Therefore, we can conclude that the proper design of a *E*/*Z* molecular switch with a high TPA cross-section ensuring the population of an excited state driving the photoisomerization is still lacking.

Here, we study different families of *E*/*Z* photoswitches in terms of TPA cross-section prediction. In particular, this work covers seven families of photoswitches (Figure 2): protonated Schiff base-like [70], Schiff base-like [22,70], oxazolone-like [71,72,73], hydantoin-like [74], and pyrrolinone-like [75] photoswitches, whose TPA properties have not been previously reported, together with azobenzene and stilbene, for which diverse experimental and computational studies exist. Although the synergy between computational and experimental TPA studies is still less ripe than for OPA studies, we consider that both points of view are essential to shed light into the rationalization of TPA cross section and their control through specific chemical modifications, at least from a qualitative perspective. For this reason, we also performed some experimental TPA spectra to support the computational findings. Thus, in this contribution we present our efforts in the computational and experimental exploration of TPA properties of well-known photoswitches which have been previously used under OPA conditions. By using an established set of switches, we could be able to compare the properties and expected applicability of these molecules under very different reactions conditions. In addition, we will use these compounds to benchmark our computational protocol and aid in the design of new molecules with improved TPA properties.

## 2. Results and Discussion

First, a screening of the TPA properties of the different proposed photoswitches was performed, including both *E* and *Z* isomers. The results are shown in Table 1, and refer to the basic core for each family of compounds, corresponding to the structures shown in Figure 2. Both isomers can, in principle, be activated by TPA, ensuring both *E* → *Z* and *Z* → *E* photochemical conversions, although the small difference between the required excitation energies could make difficult to separate the two photoprocesses in practice. However, in some cases the brightest vertical transition of the *E* and *Z* forms is separated by more than 0.3 eV. This involves the S_0_ → S_1_ transition for the protonated Schiff base-like photoswitch, the S_0_ → S_2_ transition for azobenzene, and the S_0_ → S_3_ for stilbene and the protonated Schiff base-like photoswitch.

We would like to stress that vertical excitations to the three lowest-energy singlet excited states (S_0_ → S_1_, S_0_ → S_2_ and S_0_ → S_3_) were taken into account. Indeed, our goal is to red-shift the absorption and, if relatively higher lying excited states lead to higher cross-sections, TPA may still take place in the IR window, making the possible excitation to S_n_ states, with *n* > 1, attractive especially if the TPA cross-section is considerably higher. Nevertheless, depending on the chromophore, the population of some excited states, although relevant from the point of view of TPA energy and cross-section, could result in the formation of byproducts, or in a reduction (if not a complete quenching) of the switching capability.

Usually, the S_1_ excited state is the one directly involved in the ultrafast isomerization, such as in the case of the ^1^(n,π*) state for azobenzene, and especially the highly celebrated ^1^(π,π*) state in retinal-like switches. Hence, we will first consider S_0_ → S_1_ vertical transitions. 

As can be seen, the TPA cross-section for the S_0_ → S_1_ transition can change by several orders of magnitude. In particular, while azobenzene and stilbene do not exceed 5 × 10^−3^ GM, phytochrome-, hydantoin- and oxazolone-like photoswitches increase the cross-section by 3 orders of magnitude (from 2 to 6 GM). On the other hand, the Schiff base-like photoswitch has strikingly different absorption properties, depending on the protonation state. Indeed, while this chromophore is intended to be used as a protonated Schiff base, it is nonetheless subjected to possible deprotonation, depending on the environmental pH. This constitutes a drawback for its experimental characterization, since, in principle, an equilibrium between the two forms (protonated and deprotonated) may be instated at an intermediate pH range. Moreover, for the deprotonated switch, the TPA cross-section is about 1 GM, while protonation leads not only to both an efficient photoisomerization [26,70], but also to the highest TPA cross-section among the studied families, increasing to 17.5 GM for the most stable isomer *E*. In order to preserve such optimal properties, deprotonation can be avoided by methylating the retinal-like switch, especially since only negligible differences were found concerning the experimental photochemical properties [70]. Moreover, methylation offers a safe procedure to ensure the experimental characterization and, eventually, the application of the switch. 

Concerning higher excited states, the S_0_ → S_2_ TPA cross-sections are lower than S_0_ → S_1_ ones, with the exception of *Z*-azobenzene and *E*-stilbene. Moreover, in the case of S_0_ → S_3_ excitation, *Z*-azobenzene and *Z*-stilbene show considerably higher TPA intensities, although the protonated Schiff base-like switch outperforms both compounds, reaching 113 (*E*) and 156 (*Z*) GM.

In an attempt to rationalize the results shown in Table 1, we have analyzed the molecular orbitals involved in the electronic transitions for the most used compounds: azobenzene and stilbene. Both photoswitches share the same π-conjugation length, whilst differences are noted in the type of intramolecular charge transfer. In particular, the corresponding *E* isomers, although energetically more stable at the ground state, present low TPA cross-sections, mainly due to the dominance of locally excited states. On the other hand, the *Z* isomers show a more complex behavior (Figure 3): S_1_ corresponds to a locally excited n,π* transition for azobenzene and to a locally excited π,π* transition for stilbene. As expected, these TPA transitions result in low cross-section values. On the other hand, the higher lying S_0_ → S_2,3_ transitions show partial charge transfer from the lateral benzylidene moieties to the central double bond (D-π-A-π-D), or the opposite pathways (A-π-D-π-A). Interestingly, the highest cross-sections are due to D-π-A-π-D transfer, while A-π-D-π-A transfer is comparable to locally excited states (0.14 GM, S_2_ of *Z*-stilbene).

Although it is usually believed that centrosymmetric topologies (in this case D-π-A-π-D and A-π-D-π-A transitions) do guarantee an increase of TPA cross-sections, we show that only when the electronic transfer is directed toward the core of the switch (i.e., D-π-A-π-D) we do observe such an increase. In particular, this can be explained in terms of the electron-deficiency of the core compared to the lateral moieties, as previously suggested [69]. 

Differently from azobenzene and stilbene, the other proposed chromophores have roughly similar TPA cross-section values for the *E* and *Z* isomer, regardless the excited state (see Table 1), suggesting a similar electronic nature. Hence, in these cases we have analyzed the molecular orbitals for the most stable *E* forms, discovering the same pattern for all cases (Appendix A), apart from the protonated Schiff-base like switch. The S_1_ and S_3_ state can be described as charge transfer states due to electronic push-pull effect (D-π-A), giving rise to a non-negligible TPA cross-section (ca. 1.0 to 4.5 GM), while S_2_ state corresponds to a dark n,π* state of partial locally excited character (ca. 0.0–0.1 GM). 

Instead, the protonated Schiff-base like switch shows a completely different behavior (Figure 4): the S_1_ and S_3_ states, both of D-π-A-π-D character, are characterized by high TPA cross-sections (17.5 and 113 GM, respectively), while S_2_ is a D-π-A state presenting a moderate TPA efficiency, 1.12 GM. As already concluded elsewhere [69], we could explain the TPA cross-section tendencies in terms of transition moments: since the D-π-A-π-D system is effectively polarized by both parts of the optical field cycle, its transition dipole moment will be larger than the corresponding D-π-A system, since this last is easily polarized in only one direction. Although this simple qualitative consideration can be considered a useful rule of thumb, the related TPA-cross section magnitude will be chromophore dependent. Indeed, in this specific system, due to the positive charge of the Schiff-base mainly located on part of the 5-membered ring, we expect that the two donor groups will not be equivalent, with the phenyl ring acting as a much strongest donor. 

To be assured that the excitation of the TPA brightest excited states still results in *E* → Z isomerization, we have performed a relaxed scan along the isomerizable C-C=C-C bond of the protonated Schiff-base-like photoswitch at the CASPT2 level, starting from each of the lowest excited state. The results are shown in Figure 5 and point toward an ultrafast photoisomerization for both S_1_ and S_2_ states (Figure 5a,b), while the excitation to S_3_ leads to a less efficient but nonetheless possible formation of the *Z* isomer. Indeed, S_1_ and S_2_ are almost isoenergetic in the Franck–Condon region, leading in both cases to a S_1_/S_0_ conical intersection. On the other hand, the S_3_ pathway (Figure 5c) shows that a small energy barrier around 5 kcal/mol must be overcome in order to reach a S_3_/S_2_ conical intersection. S_2_ then crosses S_1_, partially reverting the isomerization, before leading to the same S_1_/S_0_ crossing region as in the previous cases. Hence, it is in principle possible to activate the *E* → Z photoisomerization exciting by TPA all three excited states. 

Considering the promising results offered by the retinal-like core, we have accordingly proposed different derivatives for this structure, with the goal of increasing the TPA cross-section values when irradiating to S_1_, which is the ideal strategy for an absorption red-shifting while maintaining an ultrafast isomerization (Figure 5a). In particular, different R_1_ and R_2_ substituents were introduced, while in the parent compound, R_1_ is a phenyl group, and R_2_ a hydrogen atom (Figure 6a). One of the keys to understand the following results (Figure 6b) is that R_1_ constitutes the donor, while R_2_ participates as acceptor moiety (Figure 4b). 

In particular, we can underly three different effects:

(1) the conjugative effect. In order to increase the conjugation length of the chromophore, a phenyl group (-Ph) was introduced as first in R_2_, and then further increased with a naphthyl group in R_1_. As can be seen, both the TPA cross section and transition energy are positively affected, since the transition energy is red-shifted, and the TPA cross section increases up to 65.8 GM. It should be noted that both substituents are not planar with respect to the building block, hence the inductive effect is also partially acting together with the most prominent conjugative one.

(2) the R_1_ effect. Adding a methoxy group to the phenyl in R_1_ increases the donor character of this moiety, also resulting in a decrease of the transition energy—although to a lesser extent compared to the conjugative effect—and in an increase of the TPA cross section. While ortho and para derivatives are similar in terms of transition energy, different TPA properties are predicted for each case. However, experimental results for the *N*-methylated derivatives, as further shown in Figure 7b’,c’ and in the Supporting Information (Appendix A), indicate that both compounds display similar characteristics resulting from comparable effects along their scaffolds. Notably, when an electron withdrawing group as NO_2_ is placed in para position (–(*p*NO_2_)Ph), the TPA value is slightly decreased from 31.9 to 20.2 GM.

(3) the R_2_ effect. Two R_2_ substituents with an electronically opposite character were selected, coupled to the most effective R_1_ substituent (–(*p*OMe)Ph): the electron donating –(*p*OMe)Ph and the electron withdrawing –(*p*NO_2_)Ph. As aforementioned, R_2_ makes part of the acceptor moiety, hence the effect is reversed compared to R_1_: –(*p*OMe)Ph decreases the TPA cross section, while –(*p*NO_2_)Ph increases it, reaching the maximum value of 70.2 GM. The transition energy slightly decreases in both cases, mainly due to an increase of the conjugation length.

As mentioned above, experiments were performed to confirm the trends observed by molecular modeling. For these studies, we selected the Schiff base-like photoswitch model (Figure 6a), with a Ph group as R_2_ in all the cases. More concretely, we examined the features of the corresponding protonated and/or methylated forms, as represented in Figure 7 for the compounds bearing as R_1_ (a) –(1-Naphthyl), (b) –(*o*OMe)Ph, and (c’) –(*p*OMe)Ph, respectively. The corresponding data are summarized in Appendix A.

From these results we can conclude that both quaternized forms are expected to affect to a similar extent the optical properties, as described before for rhodopsin-based molecular switches [70]. However, it should be noted that, as previously stated, the protonation process depends on the pH value, and therefore might not occur in a quantitative fashion, whereas the methylation ensures the experimental characterization of the cationic quaternized photoswitch. Indeed, as it can be seen in Figure 7 (a vs. a’, and b vs. b’), the OPA spectra are almost superimposable in the absorption region of interest, while the TPA spectra are qualitatively similar in shape, with differences that could arise from the fact that the protonated form is in equilibrium with the deprotonated one. That is, both species are contributing to the spectrum up to some extent. This behavior is precisely well illustrated by the 1-Naphthyl derivative, where methylated analogue yields the highest TPA cross section within the examined compounds; this, in turn, reflects an improved π-conjugated system along the backbone. In contrast, and according to what it is anticipated for these quaternized forms, similar TPA properties are determined in the case of –(*o*OMe)Ph derivatives. Based on this reasoning, we decided to investigate only the properties of the methylated –(*p*OMe)Ph analogue. When comparing the TPA properties of the methylated analogues of these positional isomers (Figure 7b’,c’), it can be further confirmed that the position of the methoxy donor group within the scaffold has minimum influence on the optical properties, i.e., the values determined for these derivatives are very close to each other, ca. 15 GM for both cases (see Appendix A).

According to Table 1, the TPA spectrum of the protonated Schiff base-like photoswitch is due to transitions to different closed-packed excited states. More specifically, the lowest three excited states, namely S_1_, S_2_, and S_3_, are the most important. Thus, the full study of the experimental properties of this type of compounds would require the consideration of all of them. However, before describing the experimental data, two important aspects should be noted about these electronic events: (a) transitions to S_1_ and S_2_ are extremely close in terms of energy, so these may become undistinguishable, and (b) the transition to the higher-energy S_3_ state should be more relevant, due to the more prominent D-π-A-π-D character (see Figure 4). Bearing these observations in mind, the absorption spectra were analyzed for both excitation regimes (Figure 7), being the different transitions that compose the spectra highlighted by green and blue panels. From these studies, it was found that the experimental trends are in agreement with the predicted cross-section values for the S_0_ → S_1_ transition (Figure 6): the -(1-Naphthyl) derivative has the largest cross-section within the series (Appendix A). Moreover, as predicted for the protonated Schiff base-like building block (Table 1) the S_0_ → S_3_ transition allows to reach consistently higher TPA values, in the range of hundreds of GM for the methylated 1-Naphthyl derivative, when methylated (Figure 7).

A possible application of the protonated Schiff base-like switch in optogenetics was also considered. Indeed, it was already shown experimentally that an azobenzene moiety can be designed for glutamate receptors, forming a maleimide-azobenzene-glutamate (M-A-G) compound [58]. The integration of a photoswitch allows in principle to control glutamate receptors by light. We have therefore modeled the same M-A-G structure and, for comparison, a newly designed maleimide-protonated Schiff base-glutamate (M-PSB-G) compound. As can be seen in Table 2, the M-PSB-G can, in principle, reach much higher TPA cross section values for both *E* and *Z* isomers, especially when considering the S_0_ → S_2_ vertical transition. Moreover, when comparing the excitation energies to their original chromophores (Table 1 and Figure 6), it can be concluded that all values, with the exception of the (*E*)M-A-G are red-shifted. Once again, these trends can be qualitatively explained by structural and electronic parameters: structurally, the increased conjugation length due to peptide bonding –(NH)–(C=O)– of the central chromophore with the maleimide and glutamate lateral moieties (Figure 8a) is consistently beneficial in almost all cases. Electronically, we can evince the magnitude of the TPA cross section by a close inspection of the involved molecular orbitals: for M-A-G, all excitations are of local (n,π*) and (π,π*) nature, moreover the S_0_ → S_2_ electronic transition of the *Z* isomer being located on the maleimide moiety, hence not driving photoisomerization (Appendix A). On the other hand, the higher M-PSB-G TPA cross section values can be rationalized based on the D-π-A nature of all transitions, recording the highest values for the S_0_ → S_2_ transition (Table 2 and Figure 8b,c). Indeed, while for *E* and *Z* S_0_ → S_1_ transitions (Appendix A) the donor is located next to the maleimide moiety, in the case of *E* and *Z* S_0_ → S_2_ transitions (Figure 8b,c) the donor is located next to the glutamate moiety, at the same time, especially for the *E* isomer, involving a larger charge transfer character toward the acceptor, that is the photoisomerizable C=C double bond. Indeed, the systematic dependence of the TPA cross section on the transition dipole moment, and therefore on the charge transfer character, was previously investigated, although on linear π-conjugated systems [76].

## 3. Materials and Methods

### 3.1. Theoretical Background

For a molecule under linearly polarized light (as we considered in this work), we can express the TPA probability in atomic units ($\delta_{a.u.}^{TPA}$), as follows [77]:(1)$$\delta_{a.u.}^{TPA}=6\left(S_{xx}^{2}+S_{yy}^{2}+S_{zz}^{2}\right)+8\left(S_{xy}^{2}+S_{xz}^{2}+S_{yz}^{2}\right)+4\left(S_{xx}S_{yy}+S_{xx}S_{zz}+S_{yy}S_{zz}\right)$$
where $S_{ii}$ and $S_{ij}$ are the elements of the (3 × 3) matrix, spanned over the Cartesian coordinates (*x*,*y*,*z*), defining the TPA transition dipole moment tensor from the ground (*g*) to the final (*f*) electronic state:(2)$$S_{gf}=\left(\begin{array}{ccc} S_{xx} & S_{xy} & S_{xz}\\ S_{yx} & S_{yy} & S_{yz}\\ S_{zx} & S_{yz} & S_{zz} \end{array}\right)$$

To convert the TPA probability to the TPA cross section, in atomic units ($\sigma_{a.u.}^{TPA}$), the following relation applies:(3)$$\sigma_{a.u.}^{TPA}=\frac{8\pi^{2}{\left(\alpha^{TPA}\right)}^{2}\hbar E^{2}}{e^{4}}\delta_{a.u.}^{TPA}$$
where *E* is the energy difference between ground and final states, and $\alpha^{TPA}$ is a TPA molecular constant, taking into account the fine structure.

On the other hand, to convert the TPA probability to the TPA cross section, in Göppert Mayer (GM) units ($\sigma_{GM}^{TPA}$), the following relation applies [78]:(4)$$\sigma_{GM}^{TPA}=\frac{8\pi^{2}\alpha^{TPA}a_{0}^{5}\omega^{2}}{c\Gamma }\delta_{a.u.}^{TPA}$$
where $a_{0}$ is the Bohr radius, c is the speed of light in vacuum, *Γ* is the line-shape function, and ω is the frequency of the incoming photon.

All of the necessary code is implemented in DALTON2016, that firstly calculates a symmetric $S_{gf}$ by applying quadratic response theory (the only option for TD-DFT calculations) for degenerate TPA, then calculates $\delta^{TPA}$ and finally $\sigma^{TPA}$ [79]. 

Such implemented theory allows to treat large molecular systems, although it lacks an explicit calculation of the two transition dipole moments required to excite first the electronic ground state to an intermediate state, and the intermediate state to the final excited state. 

### 3.2. Computational Strategy

Excited state calculations were performed in vacuum by applying time dependent-density functional theory (TD-DFT). Specifically, each structure was first optimized on the electronic ground state (S_0_) at the B3LYP/6-31G(d) level, followed by one- and two-photon absorption (OPA and TPA, respectively) calculations. In detail, CAM-B3LYP and M06-2X functionals were tested, together with the 6-31+G(d) basis set, for a subset of structures. Moreover, the effect of the basis set was tested on azobenzene, stilbene, and the protonated retinal-like systems, by applying additionally the 6-311++G(d,p) and cc-pVTZ basis sets. Linear and quadratic absorption responses do not show a high sensitivity to the basis set and functional (see Appendix A). Such benchmark tests allow to assume that the correct qualitative ordering and the predictable systematic errors are obtained with the CAM-B3LYP functional, that we have therefore used together with the computationally affordable 6-31+G(d) basis set.

Moreover, in previous works by Beerepoot et al. CAM-B3LYP was found to be the optimal functional to calculate TPA cross sections, when compared with the higher level 2nd order coupled cluster (CC2) method, although in general it has been shown that long-range corrected functionals significantly underestimate the calculated TPA cross sections [80,81]. The main source of the discrepancies lies in the underestimation of the excited-state dipole moment of the final state, while a minor error arises from the overestimation of the excitation energy [82].

TPA cross sections have been calculated as quadratic residue of the linear response, as it was implemented by Rizzo and coworkers [83]. All TPA cross section values are given in Göppert–Mayer (GM) units, corresponding to 10–50 cm^4^ photon^−1^, in order to facilitate the comparison with the experimental data. Atomic partial charges were extracted by natural bond orbital (NBO) analysis. 

The effect of solvent (water) was considered implicitly by applying the integral equation formulation of the polarizable continuum model (IEF-PCM)—available for both linear response [84] and quadratic response [85] calculations—on selected structures of the retinal-like photoswitches.

In order to check if the excited states of the unsubstituted retinal-like photoswitch can lead to (ultrafast) photoisomerization, it was necessary to use the ab initio multiconfigurational method CASPT2 [86], i.e., complete active space self-consistent field including perturbation theory to the second order to take into account the dynamic electron correlation on top of the CASSCF wavefunction. Indeed, a multiconfigurational approach is necessary to correctly describe the photoisomerization mechanism of retinal-inspired photoswitches, involving a conical intersection between an excited and the ground state. In more detail, full CASPT2 relaxed scans (i.e., optimizations computed by calculating the CASPT2 gradient) of the photoisomerizable C-C=C-C bond were performed, starting from the Franck–Condon structure of the *E* isomer, when irradiating S_1_, S_2_, and S_3_. An IPEA value of 0.0 [87] and an imaginary shift of 0.2 [88] were applied, in order to avoid intruder states. An active space of 10 electrons in 10 molecular orbitals was selected, thus including all π and π* orbitals. Although highly time consuming, the full CASPT2 approach was preferred to the more affordable CASPT2//CASSCF approach (i.e., optimizations computed by calculating the CASSCF gradient, followed by CASPT2 single point energy correction), due to root flipping experienced with the CASSCF description, that would hamper an adequate physical description of the system.

All DFT structures were optimized with the Gaussian 16 suite of programs [89], while TD-DFT OPA and TPA calculations were performed with the DALTON2016 code [79]. The CASPT2 calculations were carried out with the OpenMolcas package [90].

### 3.3. Experimental Section

Photophysical measurements:

Spectroscopic grade solvents were employed for all the photophysical measurements. The retinal-like compounds were dissolved in acetonitrile to a final concentration below 2.00 × 10^−5^ M. When it was necessary, 1 eq. of trifluoroacetic acid was added to protonate the Schiff -base. The alkylation of the bases was carried out as described elsewhere [70]. Experiments were performed at room temperature, with aerated solutions using 1 cm pathlength quartz cuvettes. Absorption spectra were recorded on a Cary 100 Bio UV-Vis Spectrophotometer, and emission spectra were registered on a JASCO FP-750 Spectrofluorometer. Fluorescence quantum yields (ϕ_F_) were determined according to the IUPAC reference protocol, using quinine sulphate in 0.5 M sulfuric acid as reference (ϕ_F_ = 0.55) [91]. 

Two-photon absorption (TPA) cross-sections (σ^TPA^) were determined using the TP-excited-fluorescence method [92,93]. Rhodamine B (concentration below 10^−8^ M in methanol, this was adjusted for each particular case) was employed as the reference under experimentally identical conditions, assuming fluorescence quantum yield to be independent of the one or two-photon excitation. Compound fluorescence properties were analyzed using a commercial inverted Leica SP5 MP confocal and multiphoton microscope equipped with a MaiTai Ti:Sapphire HP laser (Spectra-Physics, Inc., Milpitas, CA, USA), tunable between 700 and 1040 nm. This laser provides of a pulse width below 100 fs in this range and a repetition rate of 80 MHz. The average power used was of 2.90 W at 800 nm. Imaging was performed using a 10× Plan APO objective (NA 0.4) focused on the air/liquid boundary, enabling the simultaneous detection of sample and background fluorescence. Fluorescence emission was registered with the integrated PMT detectors, and images were recorded using a 256 × 256 pixel resolution and a scan frequency of 600 Hz. The examined volume consists of a practically flat squared area, whose dimensions were 1550 × 1550 × 0.3 µm. Emission and excitation spectra data for compound and background regions of interest (ROIs) were registered using Leica LAS AF software. Spectra were measured in a laser power regime where the fluorescence was proportional to the square of the laser excitation power and using a dynamic 10 nm wide emission detection window moving in 20 steps from 400 to 700 nm. 

## 4. Conclusions

We have studied non-linear optical properties, and particularly TPA, for different families of molecular photoswitches, with the goal of selecting the chromophore with the largest cross-section coupled with the most red-shifted absorption energy and the preservation of the switching efficiency. Such properties would indeed ensure an easier applicability of photoswitches in biological and biomedical media, since the light penetration into the body tissue is maximal in the near-IR windows. 

We have found two relevant results:

(1) among all possible topologies of the electronic transitions, the D-π-A-π-D was found to be the most effective in increasing the value of the TPA cross section;

(2) the protonated Schiff base- (retinal-) like switch clearly outperforms the other families.

We have therefore designed different retinal-like switches modifying the type of donor and acceptor groups, consistently maintaining the D-π-A-π-D arrangement, to further red-shift TPA maxima. Moreover, having found that excitation to S_3_ results in a considerable increase of the TPA cross section (113 GM), we have studied by multiconfigurational quantum chemistry the photoisomerization pathways from different excited states (S_1_, S_2_, S_3_), finding out that, in principle, the *E* → *Z* photoreaction is possible in all cases, although more favorable when irradiating S_1_ or S_2_. We have compared the computational data with experimental measurements for selected examples. A good agreement between both types of data has been found.

Lastly, we have proven that the selected protonated Schiff-base like switch could be employed in optogenetics applications: the maleimide-azobenzene-glutamate receptor was “converted” into a maleimide-protonated Schiff base-glutamate receptor, i.e., substituting the photoswitch. The results suggest an increase of the expected TPA from one to two orders of magnitude, although in this case extension of the π-conjugation results in a strong D-π-A charge transfer character, instead of a D-π-A-π-D pattern, where the acceptor moiety is conserved as in the initial building block (i.e., the C=C photoisomerizable double bond), but two different donor moieties distinguish the transition to S_1_ or S_2_, this last leading to a TPA cross section of 609.0 GM. 

Overall, our study establishes a step towards the controversial and sometimes random design of TPA absorbers. In particular, in this case we have proposed a consistent enhancement of the TPA properties of photoswitches, that are expected to ensure *E* → *Z* isomerization and photoreversion, apart from TPA absorption. Compared to most of the previous bibliography on TPA absorbers, we should highlight that, in light of our results, the design could not be limited to linear and/or centrosymmetric structures, being the nature of the electronic transition and the amount of charge transfer character the most important properties to be taken into account.

## Figures and Tables

**Figure 1 molecules-26-07379-f001:**
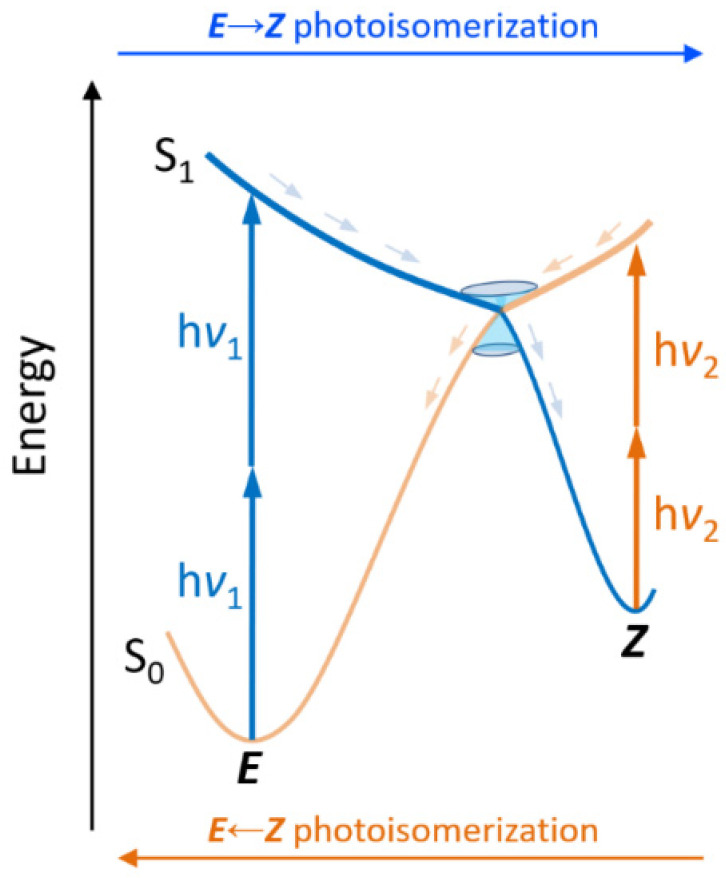
Simplified scheme of a *E*/*Z* molecular photoswitch acting by two-photon absorption, including two different frequencies: ν_1_ for *E* → *Z* (blue) and ν_2_ for *E* ← *Z* (orange) photoisomerization. After absorption to the excited state (S_1_) a conical intersection can ideally funnel the formation of the photoproduct or internal conversion to restore the initial photoisomer.

**Figure 2 molecules-26-07379-f002:**
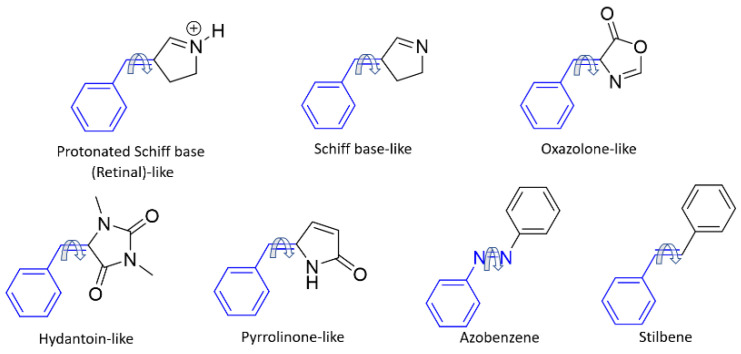
Photoswitches studied in this work. All of them have in common a benzylidene ring covalently linked to a rotatable C=C double bond, apart from azobenzene, which contains a rotatable N=N double bond (shown in blue). The most stable *E* isomers are represented.

**Figure 3 molecules-26-07379-f003:**
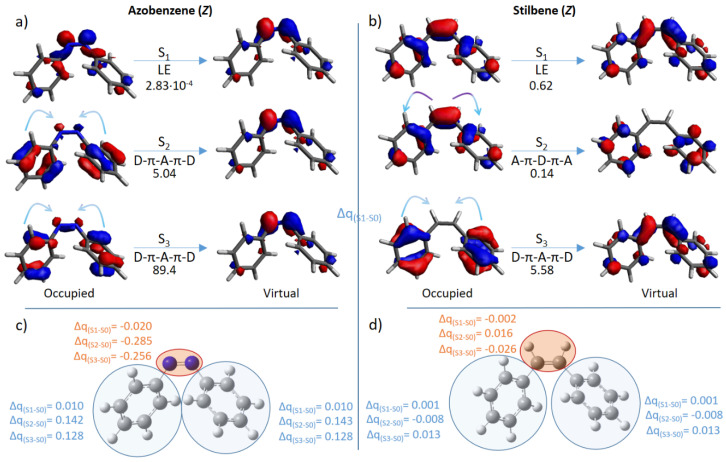
Molecular orbitals mainly involved in the lowest-in-energy optical transitions (S_0_ → S_1,2,3_) for the *Z* form of azobenzene (**a**) and stilbene (**b**). The type of electronic transfer (locally excited (LE); from the lateral moieties to the central one (D-π-A-π-D); from the middle moiety to the lateral moieties (A-π-D-π-A) and the σ^TPA^ value in GM are given below each arrow. The partial charge transfer (Δq) of groups of atoms (highlighted in light blue and orange) from each excited state to S_0_, is shown for *Z*-azobenzene (**c**) and *Z*-stilbene (**d**).

**Figure 4 molecules-26-07379-f004:**
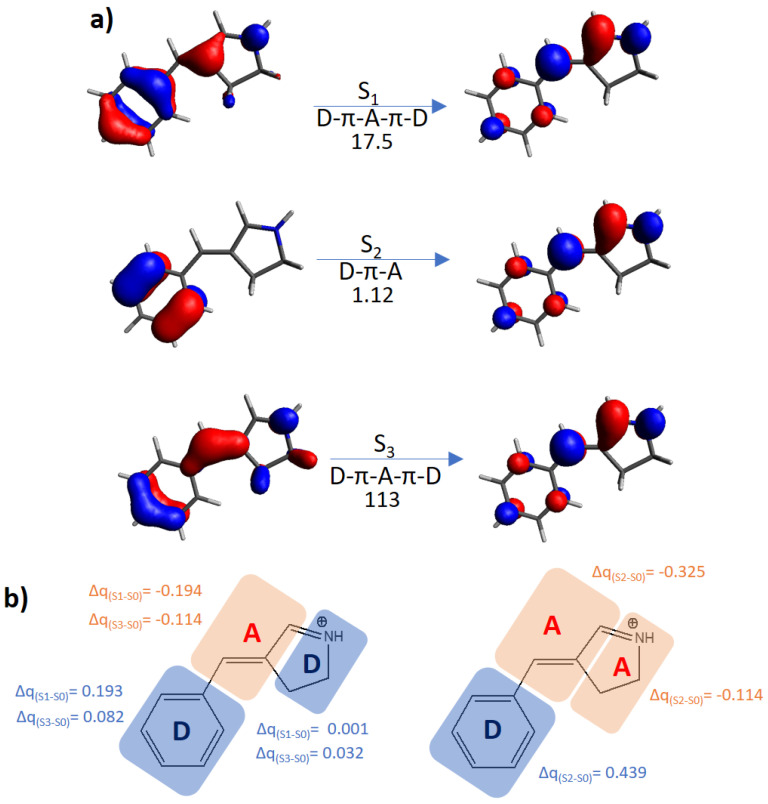
(**a**) Molecular orbitals mainly involved in the lowest-in-energy optical transitions (S_0_ → S_1,2,3_) for the *E* form of the protonated Schiff-base. The type of electronic transfer (from the lateral moieties to the central one (D-π-A-π-D); from the six-membered ring to the five-membered ring (D-π-A)) and the σ^TPA^ value in GM are given below each arrow. (**b**) Molecular moieties assigned as donor D or acceptor A. On the left, it is shown the D-π-A-π-D structure, on the right the D-π-A structure, with the relative charge transfer among excited and ground states (Δq).

**Figure 5 molecules-26-07379-f005:**
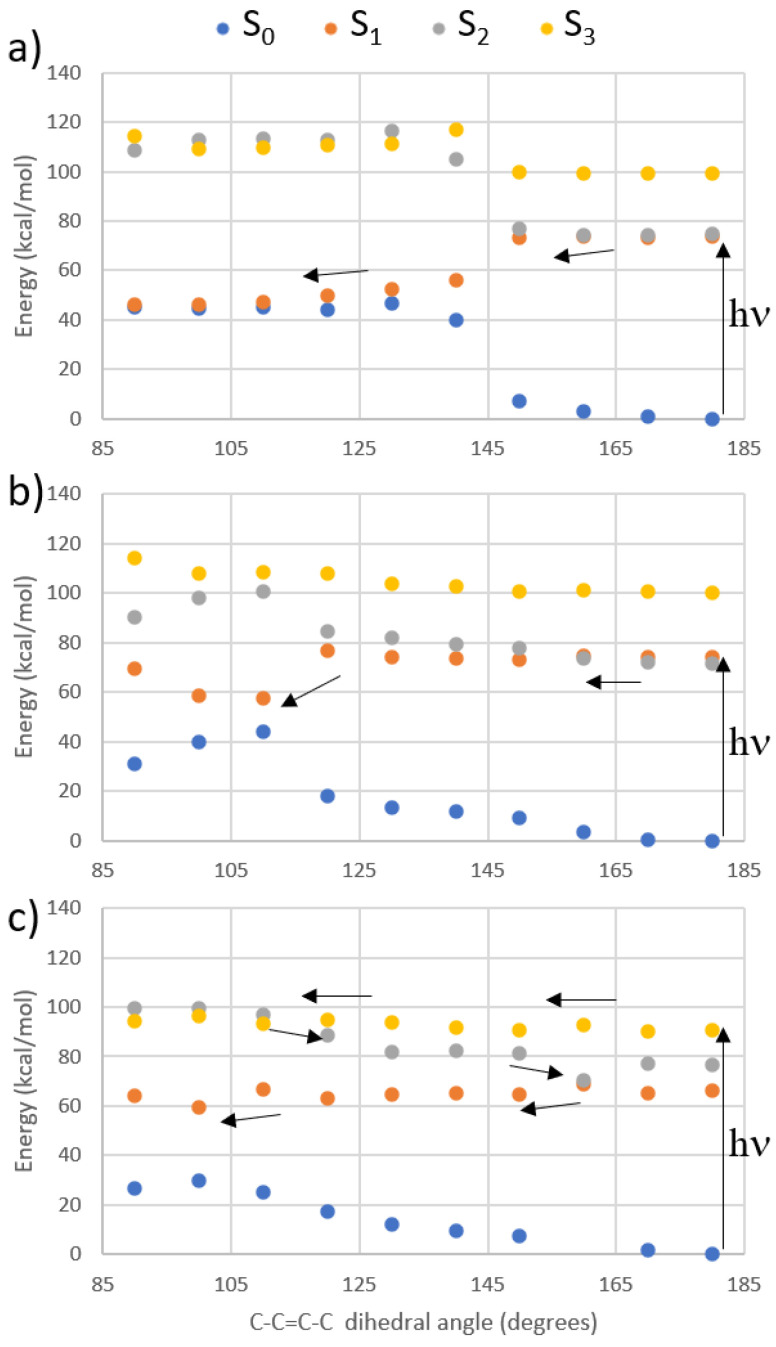
Single State (SS)-CASPT2 photoisomerization pathways when irradiating (**a**) S_1_, (**b**) S_2_, and (**c**) S_3_, shown with arrows.

**Figure 6 molecules-26-07379-f006:**
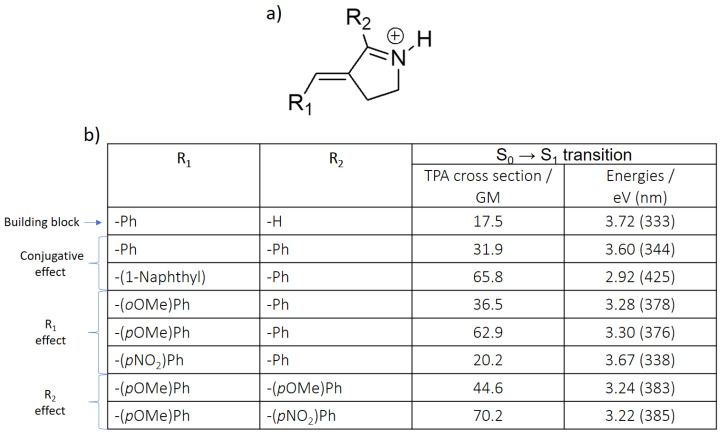
(**a**) Chemical structure of the protonated Schiff base-like derivatives. (**b**) S_0_ → S_1_ related TPA cross-section and excitation energy values of the *E* isomer, calculated at the CAM-B3LYP/6-31+G* level of theory.

**Figure 7 molecules-26-07379-f007:**
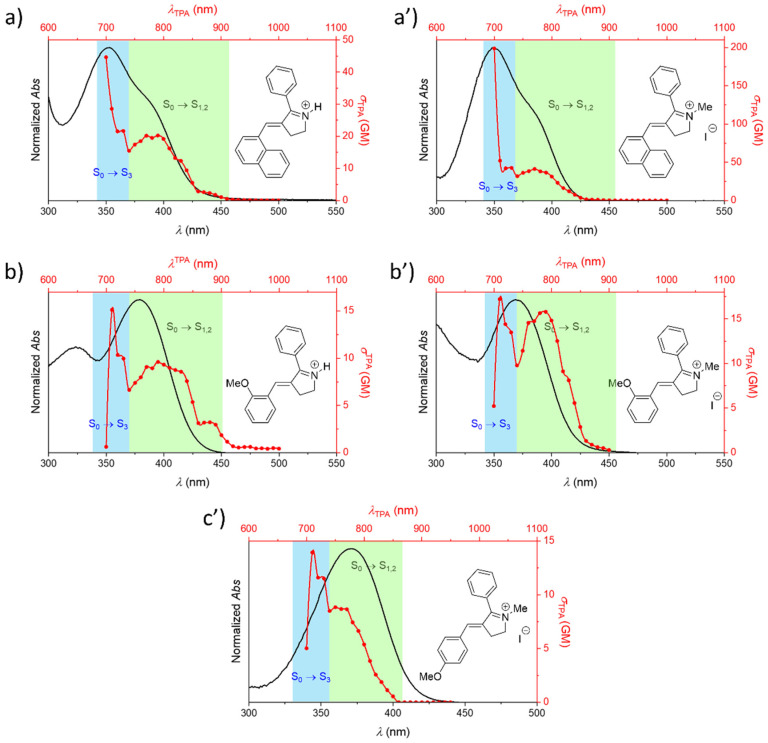
Comparison between experimental OPA (black lines) and two-photon absorption (TPA) spectra (red lines) of protonated Schiff base-like photoswitches and, when appropriate, also of their corresponding methylated analogues. The different studied derivatives are noted as (**a**) R_1_: -(1-Naphthyl), R_2_: Ph (protonated molecule); (**a’**) R_1_: -(1-Naphthyl), R_2_: Ph (methylated molecule); (**b**) R_1_: -(*o*-OMe), R_2_: Ph (protonated molecule); (**b’**) R_1_: -(*o*-OMe), R_2_: Ph (methylated molecule); and (**c’**) R_1_: -(*p*-OMe), R_2_: Ph (methylated molecule). The main electronic transitions participating in the spectra are further indicated.

**Figure 8 molecules-26-07379-f008:**
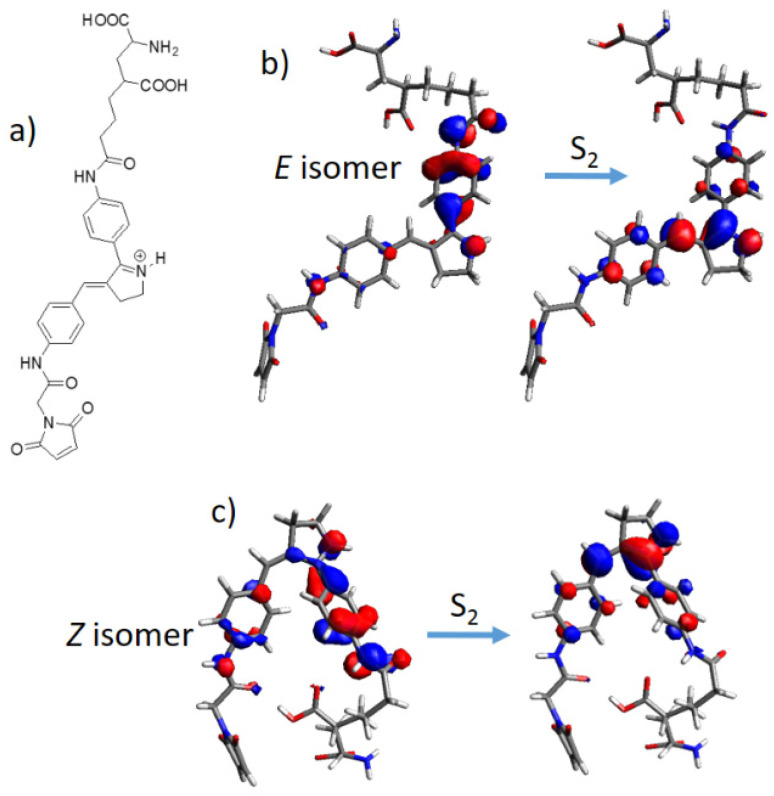
(**a**) Maleimide-protonated Schiff base-glutamate (M-PSB-G) structure; (**b**) *E* isomer and (**c**) *Z* isomer molecular orbitals involved in the S_0_ → S_2_ electronic transition.

**Table 1 molecules-26-07379-t001:** Vertical transition energies, calculated as energy difference between the electronic ground state and the selected excited state (λ), and TPA cross section values (σ^TPA^) for the different core structures of each photoswitch family studied in this work. S_0_ → S_1_, S_0_ → S_2_ and S_0_ → S_3_ vertical transitions are considered, calculated at the CAM-B3LYP/6-31+G* level of theory.

Building Block	Isomer	S_0_ → S_1_		S_0_ → S_2_		S_0_ → S_3_	
		λ/ eV (nm)	σ^TPA^/ GM	λ/ eV (nm)	σ^TPA^/ GM	λ/ eV (nm)	σ^TPA^/ GM
Protonated Schiff base-like	*E*	3.72 (333)	17.5	3.93 (315)	1.12	5.63 (220)	113
	*Z*	3.42 (363)	12.3	3.77 (329)	1.38	5.29 (234)	156
Schiff base-like	*E*	4.51 (275)	0.72	4.83 (257)	0.07	4.95 (250)	1.34
	*Z*	4.49 (276)	1.14	4.76 (260)	0.13	5.02 (247)	1.03
Oxazolone-like	*E*	3.90 (318)	2.39	4.31 (288)	6.00 × 10^−4^	4.50 (276)	1.25
	*Z*	3.96 (313)	5.42	4.33 (286)	1.00 × 10^−4^	4.57 (271)	1.57
Hydantoin-like	*E*	3.95 (314)	2.96	4.58 (271)	5.00 × 10^−4^	4.69 (264)	1.49
	*Z*	4.19 (296)	3.32	4.67 (265)	0.33	4.91 (253)	1.74
Pyrrolinone-like	*E*	3.86 (321)	4.45	4.28 (290)	0.04	4.85 (256)	1.79
	*Z*	3.93 (315)	5.90	4.32 (287)	0.12	4.84 (256)	1.96
Azobenzene	*E*	2.74 (452)	4.69 × 10^−3^	3.97 (312)	3.85 × 10^−6^	4.63 (268)	1.09 × 10^−4^
	*Z*	2.66 (466)	2.83 × 10^−4^	4.60 (270)	5.04	4.71 (263)	89.4
Stilbene	*E*	4.13 (300)	2.83 × 10^−7^	4.90 (253)	1.25 × 10^−5^	5.70 (218)	1.78 × 10^−2^
	*Z*	4.38 (283)	0.62	4.93 (251)	0.14	5.11 (243)	5.58

**Table 2 molecules-26-07379-t002:** Vertical transition energies, calculated as energy difference between the electronic ground state and the selected excited state (λ), and TPA cross section values (σ^TPA^) for two photoswitches (azobenzene, A, and protonated Schiff base-like, PSB) included as the core of the M-A-G and M-PSB-G structures. Level of theory: CAM-B3LYP/6-31+G*.

Photoswitch	Isomer	S_0_ → S_1_		S_0_ → S_2_	
		λ/ eV (nm)	σ^TPA^/ GM	λ/ eV (nm)	σ^TPA^/ GM
Protonated Schiff base-like	*E*	3.16 (392)	43.2	3.76 (330)	609.0
	*Z*	2.86 (434)	22.0	3.62 (343)	123.0
Azobenzene	*E*	2.80 (443)	8.04 × 10^−3^	3.54 (350)	2.07
	*Z*	2.60 (477)	4.57 × 10^−2^	3.88 (320)	4.40 × 10^−2^

## Data Availability

Not applicable.

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
