# Peer review of "E/Z Molecular Photoswitches Activated by Two-Photon Absorption: Comparison between Different Families"

_molecules, 2021, doi:10.3390/molecules26237379_

Round 1

Reviewer 1 Report

RECOMMENDATION: This paper is appropriate for Molecules after major revision.

This paper describes the E/Z photoswitches induced by two-photon absorption in several molecules using TDDFT. Moreover, some experimental results about the TPA spectra are reported as well. However, it contains some deficiencies outlined below that must be addressed. Thus the manuscript will only be suitable for publication after further revision.

  • The authors need to be more explicit about the manuscript's contribution to the research field because the molecules studied were exhaustively investigated in the past.
  • E/Z photoswitches molecules are very employed in optical switches, 3D optical storage, relief grating, and so on, including nonlinear optical effects, which the authors did not mention in the manuscript's introduction.
  • The experimental TPA spectra need to be moved from the SI to the manuscript.
  • It is necessary to add more details about the femtosecond laser (repetition rate, temporal duration, spatial profile, average power employed) and experimental TPA fluorescence setup.
  • A better explanation about the experimental TPA results needs to be added.
  • Many papers were published on the TPA cross-section (including excited-state dynamics and TPA cross-section values) for the retinal, retinoic acid, carotenoids, and similar molecules that the authors did not mention in the manuscript.
  • The results about the Maleimide-protonated Schiff base-glutamate are very poor. For example, the authors reported a TPA cross-section of about 609 GM for this molecule. However, such a value seems to be very high for a molecule with a weak donating and acceptor group and a small length of conjugation.  A better explanation needs to be added in the new version of the manuscript.
  • Details about the conversion from the TPA transition probability to the TPA cross-section need to be included in the new version of the manuscript.

If these changes are implemented satisfactorily, the manuscript will be suitable for publication in Molecules.

Author Response

Please see the attachment (answers to the comments and questions are written in light blue).

Reviewer 2 Report

In this manuscript, the authors investigated the cross section of the two-photon absorption (TPA) for E/Z photoisomerization. Because a molecule showing E/Z photoisomerization can be utilized as a photoswitch for several attractive fields such as smart materials, data storage, and biological applications, the enhancement of the efficiency of the two-photon absorption is important to increase the photosensitivity to upon red or NIR light and the spatial resolution of the photochemical reaction. The authors reported the TPA properties of seven different photoswitches based on azobenzene and stilbene structures. They clearly show the characteristic features of the transition by DFT calculations, and the discussion is reasonable to explain the value of cross sections. The authors also suggest the efficient molecular design of Schiff-base photoswitch for glutamate receptors. These observations will be useful and beneficial for the photochemical scientists. Therefore, the reviewer concluded this study potentially has broad interest for the readers of Molecules, and give an attractive insight for the development of photochromic studies. However, the reviewer also believe that more detailed explanations of the TPA features are required before publication. Therefore, the manuscript will be acceptable after minor revision.

(i) The authors concluded the introduction of donor and acceptor (especially D-π-A-π-D) units is important to increase the value of the cross section. However, this molecular design concept has been already known theoretically and experimentally. The point to be improved in this paper is the lack of the discussion based on the theoretical equations. The cross section for TPA has been already explained and the dipole moment is one of the important factors to increase (e.g. K. Kamada, et al. J. Phys. Chem. C, 2009, 113, 11469). Therefore, the authors should include some detailed discussions to explain why the donor-acceptor system is efficient in the main text.

(ii) Generally, the molecular design of D-π-A-π-D (or A-π-D-π-A) will be applied to increase the TPA cross section of symmetric and linear molecules. However, the Z-isomers discussed in this article are bend structures. The authors should give more detail explanation why the molecular design of D-π-A-π-D is also efficient for the (nonlinear) Z-isomers.

(iii) Lines 222-224. The authors concluded that the TPA transitions to the S1 state result in low cross-section values in both cases of azobenzene and stilbene. However, the values between those of azobenzene and stilbene are about 1000 times farther apart. What is the origin of this difference? Please discuss about it in the main text.

(iv) Lines 276-278. It seems that the reason why the shiff-base molecule shows the larger cross-section than those of the other molecules is unclear through the manuscript. Please explain that more clearly.

(v) minor point, spelling, “naphthile” → “naphthyl” (line 403, 445)

Author Response

Please see the attachment (answers to comments and questions are written in light blue).

Round 2

Reviewer 1 Report

After careful revision, I recommend this manuscript for publication in Molecules.